# Mechanistic understanding of human SLFN11

Felix J. Metzner[1,2], Simon J. Wenzl[1,2], Michael Kugler ®[1,2], Stefan Krebs[1], Karl-Peter Hopfner ®[1] & Katja Lammens ®[1] ✉

Schlafen 11 (SLFN11) is an interferon-inducible antiviral restriction factor with tRNA endoribonuclease and DNA binding functions. It is recruited to stalled replication forks in response to replication stress and inhibits replication of certain viruses such as the human immunodeficiency virus 1 (HIV-1) by modulating the tRNA pool. SLFN11 has been identified as a predictive biomarker in cancer, as its expression correlates with a beneficial response to DNA damage inducing anticancer drugs. However, the mechanism and interdependence of these two functions are largely unknown. Here, we present cryo-electron microscopy (cryo-EM) structures of human SLFN11 in its dimeric apoenzyme state, bound to tRNA and in complex with single-strand DNA. Full-length SLFN11 neither hydrolyses nor binds ATP and the helicase domain appears in an autoinhibited state. Together with biochemical and structure guided mutagenesis studies, our data give detailed insights into the mechanism of endoribonuclease activity as well as suggestions on how SLFN11 may block stressed replication forks.

Human Schlafen 11 (SLFN11) acts as a potent restriction factor of certain retroviruses, such as the human immunodeficiency virus 1 (HIV-1)[1]. SLFN11 binds tRNAs and counteracts changes in the tRNA repertoire induced by HIV-1 infections[1]. This results in the inhibition of viral protein expression in a codon-usage-dependent manner[2]. In addition to retroviruses, SLFN11 impairs the replication of DNA viruses like human cytomegalovirus (HCMV)[3] as well as positive-strand RNA Flaviviruses like West Nile virus (WNV), Dengue virus (DENV), and Zika virus (ZIKV)[4]. Hence, SLFN11 is an important antiviral factor that targets different types of viruses, offering therapeutic potential[5].

SLFN11 expression levels have been reported to show a strong positive correlation with the sensitivity of tumour cells to DNA-damaging agents (DDAs). The downregulation of certain tRNAs by SLFN11 inhibits the translation of the central DNA damage response (DDR) proteins ataxia-telangiectasia mutated (ATM) and Rad3-related protein (ATR)[6]. However, SLFN11 is also directly recruited to sites of DNA damage and stalled replication forks in response to replication stress induced by DDAs[7]. It interacts with Replication Protein A (RPA1)[7,8] and minichromosome-maintenance 3 (MCM3) at replication foci and selectively blocks fork progression by chromatin opening in the vicinity of replication initiation sites[7,9].

SLFN11 can serve as a biomarker to predict the response to platinum-based DDAs[7,10-12], topoisomerase inhibitors[9,13-17], poly(-ADP-ribose) polymerase (PARP) inhibitors[18-20] and DNA synthesis inhibitors[14,21]. As these drugs lead to replication fork stalling and cell cycle checkpoint activation, replication stress appears to be the common mechanism by which SLFN11 sensitizes cancer cells to DDAs[22].

While much has been learned about the biological and medical relevance of SLFN11, the mechanism of how the protein achieves its endoribonucleolytic and replication fork binding functions is still not well understood. Although crystal structures of the N-terminal domains of SLFN5 and rat rSLFN13 as well as cryo-EM structures of full-length SLFN5 and SLFN12 have been solved by us and others, information about substrate recognition and processing as well as the C-terminal helicase domain is still missing[23-26].

In this work, we provide the cryo-EM structures of the full-length SLFN11 apoenzyme and in complex with single-strand DNA (ssDNA) as well as tRNA. SLFN11 forms a homodimer with the helicase domains adopting an autoinhibited conformation. Guided by the structures, we predict amino acid exchanges that selectively diminish ssDNA binding and endonucleolytic activity, which we

[1]Gene Center and Department of Biochemistry, Ludwig-Maximilians-Universität München, Feodor-Lynen Straße 25, 81377 Munich, Germany. [2]These authors contributed equally: Felix J. Metzner, Simon J. Wenzl, Michael Kugler. ✉e-mail: klammens@genzentrum.lmu.de

verify by biochemical analysis. Together, the data give detailed insights into substrate recognition and processing by SLFN11 as well as its regulation by phosphorylation. The mode of ssDNA binding by dimeric SLFN11 suggests a mechanism how the protein might block stalled replication forks. Taken together, our manuscript describes a new avenue to a mechanistic understanding of Schlafen proteins and specifically how SLFN11 acts as a double-edged sword with two functions.

## Results and discussion

### Overall structure of dimeric SLFN11 apoenzyme

SLFN11 consists of an N-terminal endonuclease domain (residues 1–353), termed Slfn core domain, followed by a linker domain (residues 354–576) and a C-terminal domain with homology to superfamily I (SF I) DNA/RNA helicases (residues 577–901) (Fig. 1a). To provide a structural basis for the nuclease and ATPase functions, we used cryo-EM to solve the structure of full-length human SLFN11 (Fig. 1b). We recorded

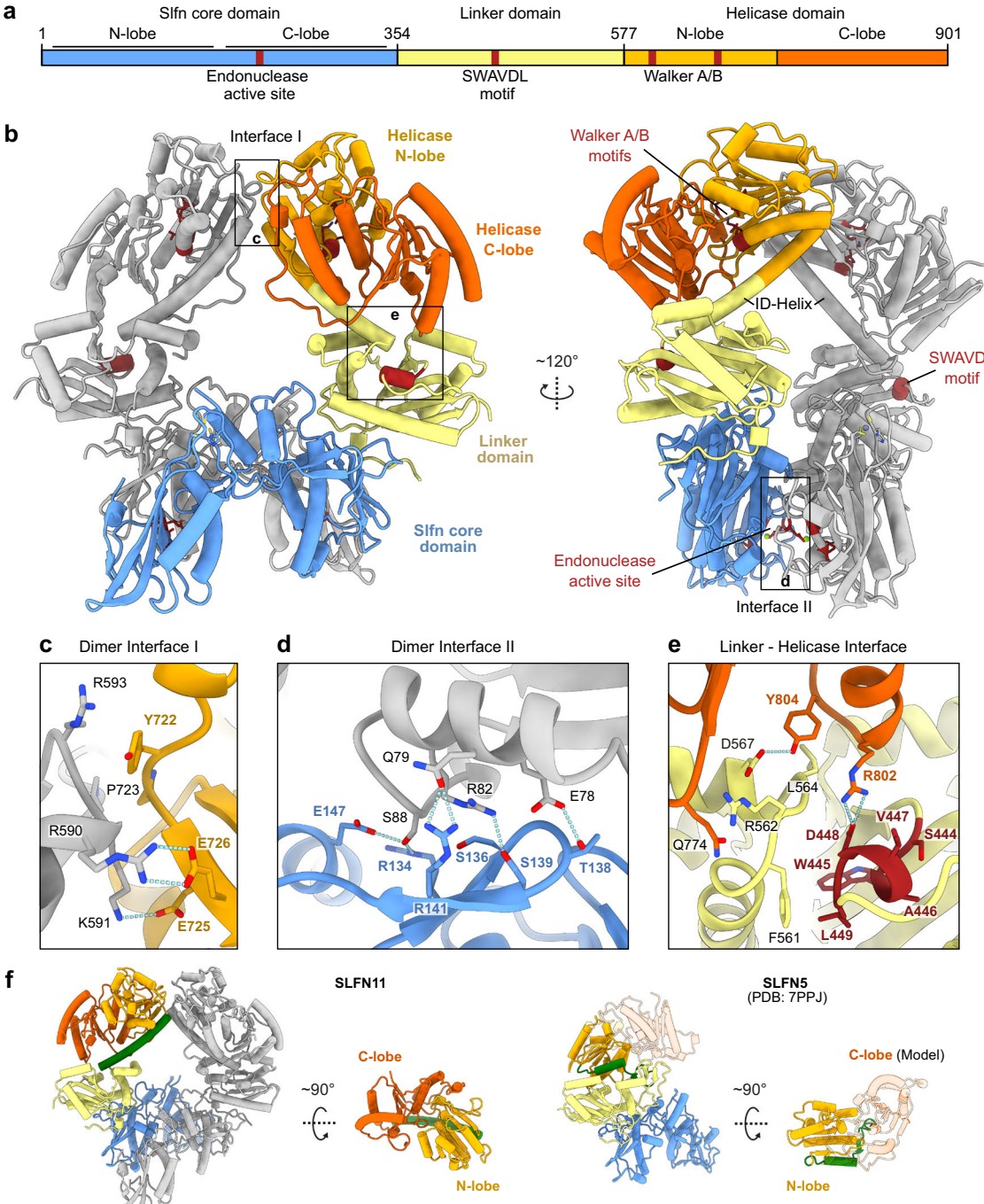

**Fig. 1 | Structure of full-length human SLFN11. a** Domain architecture of SLFN11 with indicated key functional features. **b** Ribbon representation of the full-length human SLFN11 dimer with highlighted structural features in dark red. **c** Detailed view of the dimer interface I between helicase domains. **d** Detailed view of the dimer interface II between Slfn core domains. **e** Detailed view of the linker-helicase interface. SWAVDL motif and corresponding residues are coloured in dark red. **f** Structural differences of SLFN11 and SLFN5 (PDB code 7PPJ) in the conformation of the helicase domains. The SLFN11 ID-helix is depicted in green. The position of the missing C-lobe of the helicase domain of SLFN5 is depicted as a transparent cartoon model based on the AlphaFold model[42].

datasets of SLFN11[wt] and SLFN11[E209A] and 2D classifications yielded classes of monomeric as well as C2 symmetric dimeric SLFN11. 3D reconstruction of the SLFN11[wt] dimer resulted in a map with a global resolution of 2.86 Å, allowing model building of residues 7–899 (Fig. 1b). Furthermore, monomeric and dimeric SLFN11[E209A] were reconstructed from two different datasets, resulting in maps with global resolutions of 4.0 Å and 3.25 Å, respectively (Supplementary Fig. 1a). Regarding a single protomer, the Slfn core domain forms a horseshoe-like shape as previously described for other Schlafen family members[23–26]. A zinc ion is coordinated by residues H285, C287, C321 and C322 forming a zinc finger motif (Supplementary Fig. 1b).

The SLFN11 dimer exhibits a ring-like structure with the individual protomers interacting via two interfaces (Fig. 1b–d). Interface I is located between the N-lobes of the helicase domains and is stabilized by several salt bridges (R590 to E726 and K591 to E725) (Fig. 1c). Interface II is located between the Slfn core domains that interact in a head-to-tail orientation and is formed by mostly polar and charged residues (E78, Q79, R82, S88, R134, T138, S139, R141, E147) (Fig. 1d). A related dimer interface was reported for SLFN12 which is stabilized in its dimeric form by small molecule-induced PDE3A binding[25,26] (Supplementary Fig. 1c).

In solution, two peaks corresponding to the molecular weights of monomeric (104 kDa) and dimeric (208 kDa) SLFN11 were detected by mass photometry analysis (Supplementary Fig. 1d). In line with the polar nature of the dimer interfaces, the equilibrium between monomer and dimer is highly salt-sensitive (Supplementary Fig. 1d).

The linker domain, harbouring the conserved SWAVDL motif, interacts with the helicase C-lobe (Fig. 1e) and connects to the helicase domain via a long α-helix that we will refer to as the inter-domain (ID)-helix (Fig. 1f). Notably, the structure of human SLFN5 revealed a strikingly different conformation of this region, despite a high sequence conservation[23] (Fig. 1f and Supplementary Fig. 2). Due to the different folds, the helicase N-lobes of SLFN5 and SLFN11 adopt relative conformations which are rotated by approximately 180° to each other with regard to the respective linker domains. While monomeric SLFN11 exhibits the same overall conformation as the protomers in the dimer, the Slfn core C-lobe is not visible in the map of monomeric SLFN11 (Supplementary Fig. 1a). This suggests conformational flexibility and indicates that dimerization is needed to stabilize the C-lobes of the Slfn core domains in a defined conformation.

## Mechanism of the SLFN11 endoribonuclease activity

SLFN11 has been reported to cleave type II tRNAs, leading to translational inhibition in a codon usage-dependent manner[6]. By modulation of the tRNA pool, SLFN11 can inhibit viral protein translation during e.g. HIV infections or translation of human ATM and ATR in response to stalled replication forks[1,6]. In the dimer structure, the lined-up Slfn core domains form a central channel. The endonuclease active sites are located in this central groove in close proximity to interface II (Fig. 2). To identify the binding mode to tRNA, we structurally analysed SLFN11 in the presence of a mixture of different yeast tRNAs. 3D

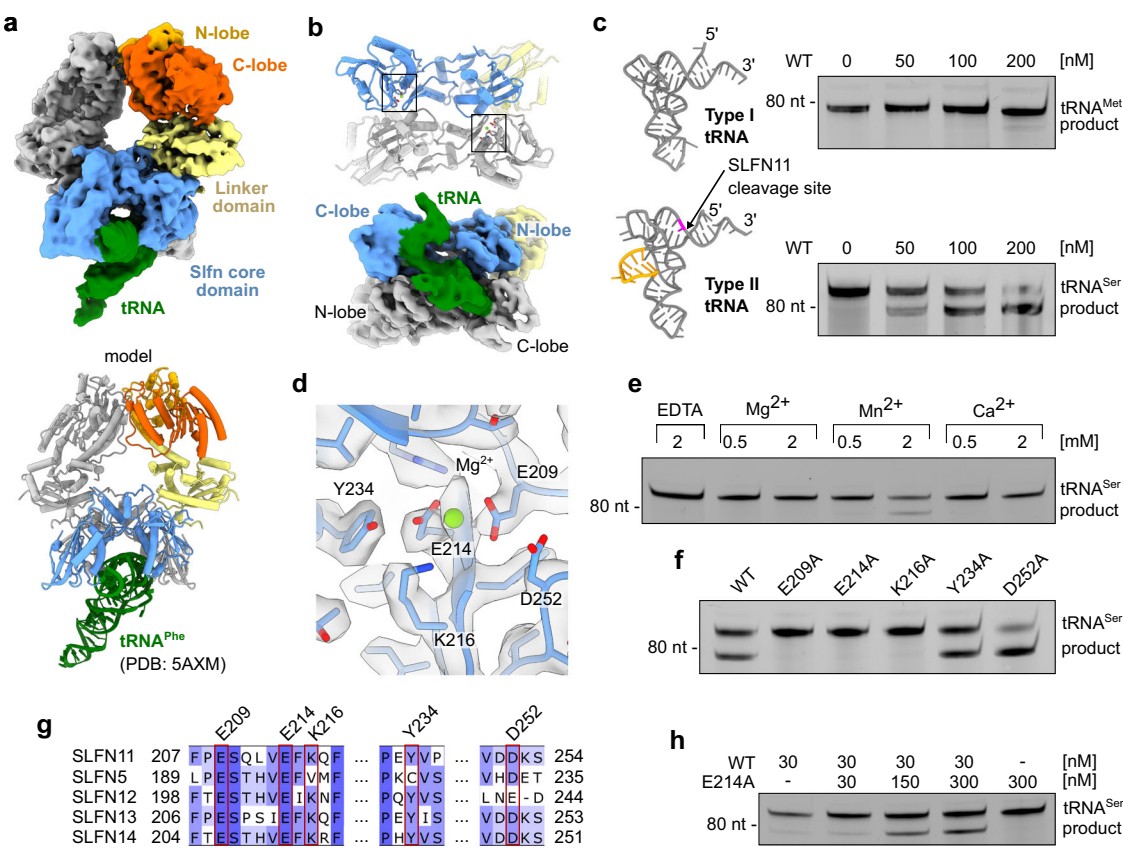

**Fig. 2 | SLFN11 tRNA recognition and nuclease activity. a** Cryo-EM density map for SLFN11 bound to tRNA with proposed structural model of docked yeast tRNA[Phe] (PDB code 5AXM). **b** Bottom view of Slfn core domains with highlighted nuclease active sites. Cryo-EM density map for tRNA bound between N- and C-lobes of Slfn core domains. **c** Structural models of type I (tRNA[Met], PDB code 2FMT) and type II (tRNA[Ser], model created by RNAComposer[53, 54]). Variable loop of type II tRNA is coloured in orange and SLFN11 cleavage site in pink (nt 75). SLFN11 endonuclease activity towards respective tRNAs demonstrated in nuclease assays. **d** Close-up view of the nuclease active site with active site residues (E209 and E214)

coordinating a magnesium ion. **e** Metal ion dependency of SLFN11 endonuclease activity examined by nuclease assay. **f** Effect of mutations of nuclease active site residues on cleavage of tRNA[Ser] monitored by nuclease assay. **g** Multiple sequence alignment of nuclease active site residues of human Schlafen family members with mutated residues highlighted in red. **h** Characterization of dimerization-induced nuclease activity of SLFN11 monitored by nuclease assay. Experiments in **c, e, f, h** were performed in duplicates. One representative replicate is shown. Source data for **c, e, f, h** are provided as a Source Data file.

reconstruction showed additional density in the groove between the Slfn core domain N- and C-lobes, fitting one tRNA molecule (Fig. 2a). This shows that a single tRNA molecule is bound and cleaved by the SLFN11 dimer. Two conserved positively charged patches, previously described to contribute to efficient tRNA cleavage by rSLFN13, are in close proximity to the tRNA[24] (Supplementary Fig. 3a). Although the resolution of the map did not allow for detailed tRNA model building, the density indicates that substrate recognition occurs in the Slfn core N- and C-lobes of both protomers of the dimer (Fig. 2b).

We analysed binding of SLFN11 to type I (tRNA$^{Met}$) and type II (tRNA$^{Ser}$) tRNAs in vitro, showing that it binds both tRNAs (Supplementary Fig. 3b). However, tRNA$^{Ser}$ is more efficiently cleaved compared to tRNA$^{Met}$ (Fig. 2c). Type I and type II tRNAs differ in the presence of a variable loop that is present in type II but not in type I tRNAs (Fig. 2c). The observed cleavage pattern places the cutting site approximately 10 nucleotides from the 3′ end, between the acceptor stem and the T-loop (Fig. 2c and Supplementary Fig. 3c, d). To precisely map the SLFN11 endonucleolytic cutting site, we sequenced tRNA$^{Ser}$ cleavage products (Supplementary Fig. 3e). SLFN11 cleaves tRNA$^{Ser}$ mainly at a single cutting site, positioned 10 nucleotides from the 3′ end, between nucleotides 75 and 76. This is in line with the cleavage pattern observed for rSLFN13[24]. In the cryo-EM map, SLFN11 interacts with the acceptor stem and T-loop of the tRNA. It is positioned in close proximity to both endonuclease active sites, approximately 10 and 20 nucleotides from the 3′ end of the tRNA, respectively. However, based on the endonuclease activity assays and the sequencing results, only the active site that is closer to the 3′ end is cleavage proficient. The variable loop which is not clearly visible in our density, could be involved in binding, since it would be positioned in proximity to the Slfn core domain. Hence, the specificity of the enzyme is presumably determined at several different recognition sites.

In the apoenzyme structure additional density for a metal ion appears at the proposed nuclease active site which is coordinated by residues E209 and E214 (Fig. 2d). In cleavage reactions, addition of $Mn^{2+}$ resulted in endonuclease activity, while $Mg^{2+}$ and $Ca^{2+}$ did not stimulate tRNA cleavage (Fig. 2e). Density for a metal ion could not be observed in the cryo-EM map of mutant SLFN11$^{E209A}$ (Supplementary Fig. 3f). To clarify, which residues contribute to the endonuclease activity, we introduced point mutations around the proposed nuclease active site. Mutation of the ion coordinating residues (E209A, E214A) abolished the nuclease activity completely (Fig. 2f). This is in accordance with the active site residues identified for rSLFN13[24]. In contrast, mutation D252A, the third residue of the proposed three carboxylate triad, resulted in a slight increase in nuclease activity. Mutation Y234A showed wild type-like activity, while K216A rendered the nuclease inactive (Fig. 2f). The essential residues E209, E214 and K216 are conserved among RNA cleavage proficient human Schlafen proteins (SLFN11, SLFN12, SLFN13, and SLFN14) (Fig. 2g). Schlafen family members lacking one of these three active site residues, like SLFN5[23] or mouse mSLFN2[27], have been shown to be endonuclease deficient.

Malone et al. reported[28] that ribonuclease activity of SLFN11 is inhibited by phosphorylation of residues S219, T230 and S753. Thus, we investigated the phosphorylation status of the protein used in our studies, which was expressed in insect cells. Mass spectrometry analysis unambiguously identified tryptic peptides with unphosphorylated S219, T230 and S753 (Supplementary Fig. 4). Corresponding phosphorylated peptides remained undetected. This confirms that the unphosphorylated protein is the enzymatically active form. Residues S219 and T230 are located within the Slfn core C-lobe in close proximity to the nuclease active site and might directly influence substrate binding or cleavage[28] (Supplementary Fig. 4a). The impact of a phosphorylation at S753 which is located within the helicase domain is not obvious from a structural point of view and will be discussed subsequently (Supplementary Fig. 4b).

To biochemically verify whether SLFN11 dimerization is required for endonuclease activity, we performed an in trans complementation assay where the nuclease inactive SLFN11$^{E214A}$ was titrated to a constant and limiting concentration of SLFN11$^{wt}$. The nuclease activity was increased with the addition of SLFN11$^{E214A}$, while SLFN11$^{E214A}$ alone was inactive (Fig. 2h). This confirms that dimeric SLFN11 is the nuclease active species and that a single active site is sufficient for tRNA cleavage. A double mutant of dimer interface I located in the helicase domains (K591D, Y722A) did not influence tRNA cleavage while the introduction of a third mutation in dimer interface II (R82D) rendered the nuclease inactive (Supplementary Fig. 3g). In summary, dimerization of SLFN11 is necessary for substrate specificity and cleavage site orientation in the active site. In line with our results, it has been hypothesized that the ribonuclease activity of subgroup II Schlafen family member SLFN12 might be stimulated by dimerization[25].

## SLFN11 binding to single-strand DNA

SLFN11 is recruited to stalled replication forks and blocks them in an irreversible manner, eventually leading to cell death[7,18,29]. We found that SLFN11 binds single-strand DNA (ssDNA) with high affinity but does not bind to double-strand DNA (dsDNA) (Fig. 3a and Supplementary Fig. 5). SLFN11 binding to 50 nucleotide (nt) ssDNA resulted in a defined species in an electrophoretic mobility shift assay (EMSA) (Fig. 3a). Fluorescence anisotropy was employed to verify the binding preference for ssDNA over dsDNA and yielded an apparent $K_d$ of ~30 nM for ssDNA (Supplementary Fig. 5a). Binding of ssDNA to SLFN11 increased its inflection temperature by 7.6 °C in a thermal unfolding assay (nanoDSF) while dsDNA had no stabilizing effect (Fig. 3b).

To structurally analyse the interaction between SLFN11 and ssDNA, we solved the cryo-EM structure of SLFN11 bound to 60 nt ssDNA at 3.16 Å resolution (Fig. 3c). The overall fold is similar to the structure of the apoenzyme dimer. Both protomers bind a stretch of five nucleotides via a positively charged patch between the helicase N- and C-lobes (Fig. 3d). Since the helicase domains of the two protomers are rotated by 180° relative to each other, the ssDNA strands are pointing in opposing 5′ to 3′ directions (Fig. 3c). The majority of the interactions are located along the phosphate backbone (helicase N-lobe: N633, Q634, T650, K652, T653, R656, R674; helicase C-lobe: S853, R855, R856) (Fig. 3d). A single charge reversal mutation (K652D) abolishes the ssDNA binding completely (Fig. 3e and Supplementary Fig. 5b, c). The mutation has no effect on the ribonuclease activity confirming the correct folding of the protein (Supplementary Fig. 5d). Most DNA binding residues are conserved amongst human subgroup III Schlafen family members except for SLFN14 and with minor differences in SLFN5 and SLFN13 (Supplementary Fig. 2). The shape of the ssDNA binding groove in SLFN11 seems to sterically prevent dsDNA binding (Supplementary Fig. 5e). NanoDSF measurements of SLFN11 in the presence of dsDNA and at different buffer conditions showed no effect on the inflection temperature (Supplementary Fig. 5f–h). This indicates that SLFN11 does not bind to dsDNA.

The sigmoid shape of the fluorescence anisotropy data suggests cooperative binding of ssDNA (Fig. 3e and Supplementary Fig. 5a), consistent with the observation that both DNA binding sites are occupied in the cryo-EM structure. Mass photometry data show that the addition of ssDNA shifts the equilibrium towards the dimer state, indicating ssDNA-induced stabilization of the SLFN11 dimer (Supplementary Fig. 5i). The opposite 5′ to 3′ direction of the two bound ssDNAs suggests that the SLFN11 dimer could simultaneously bind to both single strands at a stalled replication fork and may thereby block fork progression[7,9].

It has been reported that SLFN11 can be phosphorylated at S753 and that the phospho-mimetic mutant S753D is incapable of reducing type II tRNA levels[28]. The phosphorylation site at S753 is located near the ssDNA binding groove within the helicase domain (Supplementary Fig. 6a). Thus, we mutated S753 to a phospho-mimetic aspartate.

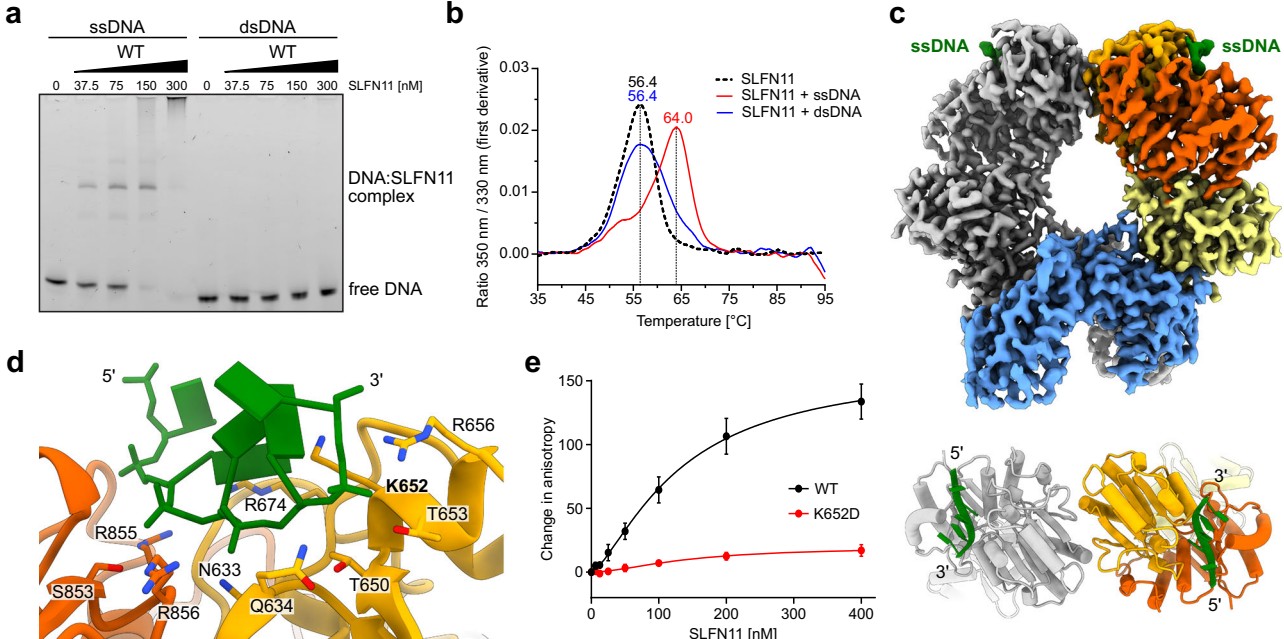

**Fig. 3 | SLFN11 binding to DNA. a** ssDNA and dsDNA binding ability of SLFN11 monitored by electrophoretic mobility shift assay. The experiment was performed in duplicates. One representative replicate is shown. **b** NanoDSF measurements of SLFN11 in the presence of ssDNA and dsDNA. **c** Cryo-EM reconstruction of the SLFN11wt dimer bound to ssDNA. Top view of ssDNA bound between the helicase N- and C-lobes of SLFN11 with indicated 5´ to 3´ direction. **d** Detailed view of the DNA binding region with labelled DNA-interacting residues. Mutated residue K652 is highlighted in bold. **e** Fluorescence anisotropy assay documenting the effect of K652D mutation on the binding of SLFN11 towards 50 nt ssDNA. The data were fit to a cooperative binding equation. Data are represented as mean values + /− SEM from four independent experiments. Source data for **a**, **b**, **e** are provided as a Source Data file.

Although there is no direct interaction between S753 and ssDNA in the cryo-EM structure, the S753D mutant is deficient in DNA binding as monitored by EMSA (Supplementary Fig. 6b, c). This hints towards a functional connection between ssDNA binding to the helicase domain and the nuclease activity of the Slfn core domain.

### SLFN11 exhibits an autoinhibited ATPase conformation

The helicase domain of SLFN11 harbours the essential Walker A and B motifs of SF I DNA/RNA helicases which usually couple ATP hydrolysis to translocation along or unwinding of DNA or RNA (Supplementary Fig. 2). Mutation of the Walker B motif has been shown to cause defects in chromatin opening activity of SLFN11 in response to replication stress[9]. Thus, we analysed nucleotide binding and hydrolysis by SLFN11. Addition of different nucleotides (ATP, ADP and ATPγS) causes only minor changes in the inflection temperature of SLFN11, indicating no interaction with the tested nucleotides (Fig. 4a). In contrast, SLFN5 shows a significant increase in unfolding temperature in presence of ATP and ATPγS (Fig. 4a).

We structurally overlaid the helicase domains of SLFN11 and DNA2, another SF I DNA/RNA helicase in its ADP-bound state (Fig. 4b). This illustrates that in SLFN11, ATP binding is sterically blocked by the ID-helix. The conserved glutamine residue of the predicted Q-motif points away from the nucleotide-binding site (Fig. 4b). In addition, we could not detect ATPase activity of SLFN11 alone or in the presence of different DNA/RNA substrates or RPA (Supplementary Fig. 7a). Together, this indicates that the ATPase domains of dimeric SLFN11 are locked in an inactive state, unable to bind ATP.

Structural comparison of SLFN11 and SLFN5 reveals large differences in the conformation of the helicase domains and linker domains (Fig. 4c). In SLFN11, the conserved residue F561 and R802 of the helicase C-lobe interact with the SWAVDL motif (amino acid 444–449) (Fig. 1e). These interactions might suggest a regulatory role of this amongst Schlafen proteins highly conserved motif. In SLFN5, the helicase N-lobe forms an interface with the linker domain and the

different conformation of the ID-region potentially opens space for a nucleotide to bind (Fig. 4b). The SLFN11 helicase domain appears to be locked in an autoinhibited conformation via the dimer interface I and the helicase-linker domain interface (Figs. 1c, e and 4c).

The monomeric cryo-EM structure adopts a similar helicase domain arrangement and ID-helix fold as the SLFN11 dimer (Supplementary Fig. 1a), indicating a second necessary step to trigger an active ATPase conformation. However, the characteristic strand separating Pin motif that is found in most SF I helicases with double strand unwinding activity is missing in SLFN11[30,31] (Supplementary Fig. 7b). Together with the absence of dsDNA binding elements (SF1A helicase domains 1B and 2B) in SLFN11, this demonstrates that the protein alone is not a strand opening helicase.

In summary, human SLFN11 is a manganese-dependent type II tRNA endoribonuclease. One tRNA molecule is bound in the positively charged groove of the SLFN11 core domain dimer and both protomers are involved in substrate recognition. Although the tRNA is in close proximity to both nuclease active sites, only one cleavage reaction occurs 10 nucleotides from the 3' end. The active site residues E209, E214 and K216 are essential for catalysis. These residues are conserved in human SLFN11, 12, 13 and 14 that were shown to be nuclease proficient[1,24,25,32,33]. Previous studies implied that helicase or ATPase activity is needed for replication fork blockage and/or chromatin opening[7]. SLFN11 on its own is not proficient in ATP hydrolysis nor ATP binding in vitro. Hence, an additional factor e.g. binding partner, modification or signal is needed to activate ATPase activity. Despite the high sequence conservation of the Schlafen family members, the subtle differences in the endonuclease active site, DNA binding groove and helicase domain lead to divergent functions within this protein family. The presented data shed light on the structural and functional attributes of human SLFN11 and how it might bind to replication forks. The structural and mutational data explain the regulatory and inhibitory effects resulting from the phosphorylation of SLFN11. The inability of the phospho-mimetic S753D mutant to cleave tRNA and to bind

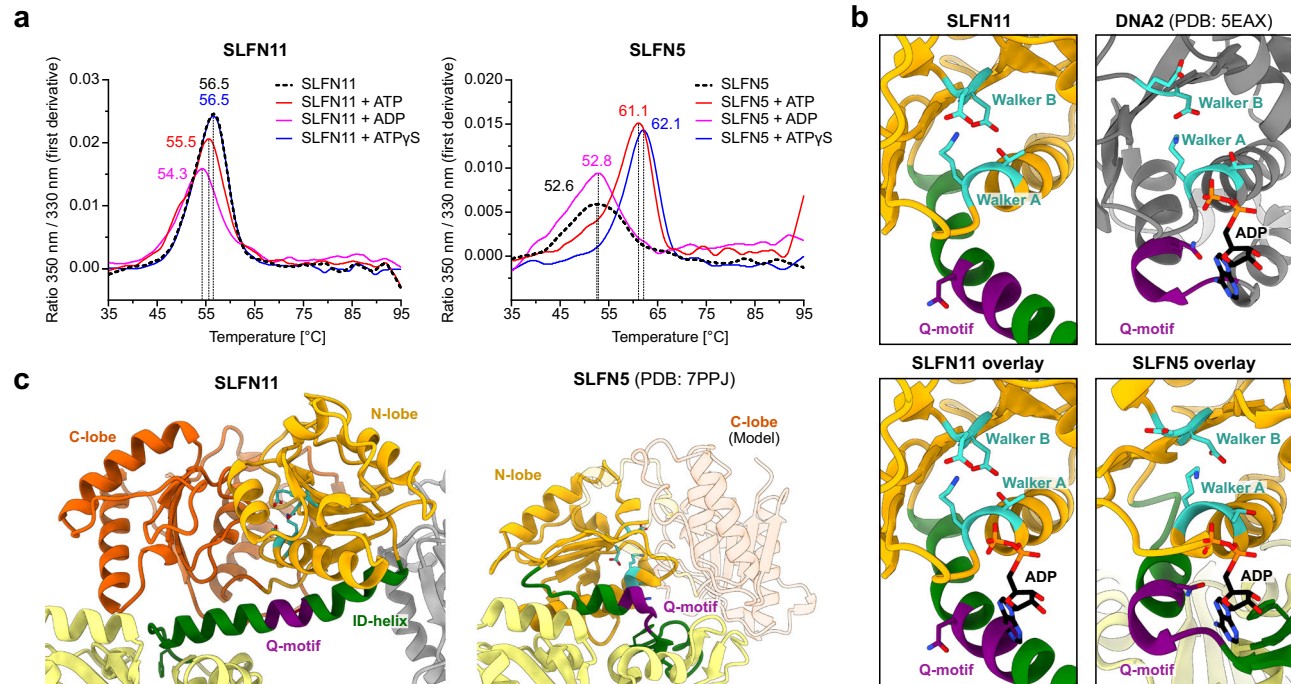

**Fig. 4 | Characterization of the SLFN11 helicase domain. a** NanoDSF measurements of SLFN11 and SLFN5 in presence of different nucleotides or without nucleotide. **b** Structural comparison of helicase domains of SLFN11, SLFN5 (PDB code 7PPJ) and DNA2 in ADP bound state (PDB code 5EAX). Walker motifs and Q-motif are highlighted in teal and purple, respectively. ADP is coloured in black. **c** Detailed view of the interdomain helix (ID-helix) conformational differences between SLFN11 and SLFN5 (PDB code 7PPJ). Walker motifs are highlighted in teal. Source data for **a** are provided as a Source Data file.

ssDNA might be a hint at a functional connection between the nuclease and helicase domains. We conclude that SLFN11 may act as a double-edged sword with two functions within one protein which might be regulatory connected. Further *in cellulo* studies are needed to uncover the whole regulatory picture of SLFN11 activation and to clarify the possible coupling or separation of these two functions. Our biochemical and structural investigation of full-length SLFN11 presents an important basis to facilitate these studies.

## Methods

### Protein expression and purification

A construct encoding for full-length SLFN11 with an N-terminal double FLAG-tag and a HRV 3 C cleavage site was purchased from GenScript. SLFN11 was cloned into the pFASTBac1 expression vector using Gibson assembly for expression in insect cells[34]. *Spodoptera frugiperda* Sf21 insect cells (Thermo Fisher) were used for virus generation. Expression was carried out in *Trichoplusia ni* High Five cells (Invitrogen) at 27 °C and 95 rpm for 72 h. After 72 h, the cells were harvested by centrifugation, resuspended in lysis buffer (50 mM Tris pH 7.5, 400 mM NaCl, 2 mM MgCl$_2$) supplemented with protease inhibitors (0.18 g l$^{-1}$ PMSF, 0.32 g l$^{-1}$ benzamidine, 1.37 mg l$^{-1}$ pepstatin A, 0.26 mg l$^{-1}$ leupeptin, 0.2 mg l$^{-1}$ chymostatin) and disrupted by sonication. The lysate was clarified by centrifugation at 30,000 g at 4 °C for 45 min and the supernatant was incubated with pre-equilibrated ANTI-FLAG M2 Affinity Gel (Sigma-Aldrich) for 90 min. The resin was washed with wash buffer (25 mM Tris pH 7.5, 250 mM NaCl, 2 mM MgCl$_2$). After washing with buffer A (25 mM Tris pH 7.5, 120 mM NaCl, 2 mM MgCl$_2$, 1 mM DTT), the protein was eluted iteratively for five times in elution buffer (buffer A supplemented with 0.2 mg ml$^{-1}$ Flag-peptide) over 60 min. The eluate was loaded onto a HiTrap Heparin HP column (GE Healthcare) and the protein was eluted by a linear salt gradient (100 % buffer A to 100 % buffer B (25 mM Tris pH 7.5, 1 M NaCl, 2 mM MgCl$_2$, 1 mM DTT) over 12 CV). The peak fractions were combined and flash frozen in liquid nitrogen. For preparation of cryo-EM samples, SLFN11 was

directly concentrated using a centrifugal filter unit (Amicon, MWCO 30 kDa). Concentrated SLFN11 was applied onto either Superdex 200 5/150 or Superose 6 increase 5/150 column (GE Healthcare) equilibrated in buffer A. Peak fractions were used for cryo-EM grid preparation.

SLFN11 mutants were prepared by site-directed mutagenesise PCR and expressed and purified as the wild-type protein. SLNF5 was expressed and purified following a similar protocol, with the difference, that the pH of the buffers was adjusted to pH 7.1[23].

RPA was cloned into the pBIG1a expression vector using the biGBac system[35]. *Spodoptera frugiperda* Sf21 insect cells (Thermo Fisher) were used for virus generation. RPA was expressed in *Trichoplusia ni* High Five cells (Invitrogen) similarly to SLFN11. Cell pellets were resuspended in lysis buffer (20 mM HEPES pH 7.8, 300 mM NaCl, 2 mM MgCl$_2$, 5 mM KCl, 0.1 mM EDTA, 4 mM imidazole) supplemented with protease inhibitors. Cells were lysed by sonication and the lysate was centrifuged at 30,000 g at 4 °C for 60 min. The supernatant was applied onto pre-equilibrated Ni$^{2+}$-NTA agarose beads (Qiagen), and incubated for 45 min. Beads were applied onto a 5 ml column (Bio-Rad) and washed with lysis buffer. RPA was eluted by adding 5 CV imidazole buffer (20 mM HEPES pH 7.8, 300 mM NaCl, 2 mM MgCl$_2$, 5 mM KCl, 0.1 mM EDTA, 350 mM imidazole). The eluate was pooled and dialyzed overnight into buffer A (20 mM HEPES pH 7.8, 100 mM NaCl, 2 mM MgCl$_2$, 5 mM KCl, 0.1 mM EDTA). The clarified protein solution was applied onto a pre-equilibrated HiTrap Heparin HP column (GE Healthcare). RPA was eluted by applying a linear NaCl gradient (10 CV) using buffer B (20 mM HEPES pH 7.8, 1 M NaCl, 2 mM MgCl$_2$, 5 mM KCl, 0.1 mM EDTA). Peak fractions were pooled and flash frozen in liquid nitrogen.

### Sample preparation and cryo-EM data acquisition

Purified SLFN11$^{wt}$ was dialyzed into dialysis buffer (20 mM HEPES pH 7.5, 60 mM NaCl, 2 mM MgCl$_2$, 1 mM DTT) and pre-incubated for 30 min with brewer's yeast tRNAs mixture (Sigma-Aldrich). The sample

was diluted in cryo-EM buffer (20 mM HEPES pH 7.5, 2 mM $MgCl_2$, 1 mM DTT) to a final concentration of 3.5 µM SLFN11 and 10 µM tRNA mixture. 4.5 µl was applied onto a glow discharged QUANTIFOIL® R2/1 + 2 nm carbon Cu200 grid. The sample was vitrified in liquid ethane using an EM GP plunge freezer (Leica, 10 °C and 90% humidity).

Freshly purified SLFN11$^{E209A}$ was diluted in cryo-EM buffer (50 mM Glycine pH 9, 200 mM NaCl, 2 mM $MgCl_2$, 1 mM DTT) to a final concentration of 5 µM. 4.5 µl was applied onto a glow discharged QUANTIFOIL® R2/1 Cu200 grid. The sample was vitrified in liquid ethane using an EM GP plunge freezer (Leica, 10 °C and 90% humidity).

Freshly purified SLFN11$^{wt}$ was pre-incubated for 30 min with 60 nt ssDNA and 1 mM ADP. The sample was diluted in cryo-EM buffer (100 mM Glycine pH 9, 2 mM $MgCl_2$, 1 mM DTT) to a final concentration of 5 µM SLFN11, 2 µM ssDNA, and 1 mM ADP. 4.5 µl was applied onto a glow discharged QUANTIFOIL® R2/1 Cu200 grid. The sample was vitrified in liquid ethane using an EM GP plunge freezer (Leica, 10 °C and 90% humidity).

## Cryo-EM data collection

Cryo-EM data were collected using an FEI Titan Krios G3 transmission electron microscope (300 kV) equipped with a GIF quantum energy filter (slit width 20 eV) and a Gatan K2 Summit direct electron detector (software used: EPU 2.12.1.278REL, TEM User interface Titan 2.15.4, Digital Micrograph 3.22.1461.0). For the structure of SLFN11$^{wt}$ and SLFN11$^{wt}$ bound to tRNA 8,569 movies were collected with a total electron dose of 49.65 e$^-$ Å$^{-2}$, fractionated into 40 movie frames over 8 s. For the structure determination of SLFN11$^{E209A}$ dimer 7,078 movies were collected (of which 4,420 movies at a tilt angle of 25°) with a total electron dose of 44.03 e$^-$ Å$^{-2}$, fractionated into 40 movie frames over 8 s. For the structure of SLFN11$^{E209A}$ monomer 3,212 movies were collected with a total electron dose of 43.58 e$^-$ Å$^{-2}$, fractionated into 40 movie frames over 8 s. For the structure determination of SLFN11$^{wt}$ bound to ssDNA 6,419 movies were collected (of which 4,088 movies at a tilt angle of 25°) with a total electron dose of 43.33 e$^-$ Å$^{-2}$, fractionated into 40 movie frames over 8 s. All datasets were collected with defocus values ranging from -1.1 to -2.9 µm and a pixel size of 1.046 Å.

## Cryo-EM image processing

Movie frames were motion corrected using MotionCor2 1.4.5[36]. All subsequent cryo-EM data processing steps were carried out using cryoSPARC 3.3.1[37] and the resolutions reported here are calculated based on the gold-standard Fourier shell correlation criterion (FSC = 0.143). In total, five datasets were processed. The CTF parameters of the datasets were determined using patch CTF estimation (multi). The exact processing schemes are depicted in Supplementary Fig. 8–10. The data collection and refinement statistics are summarized in Supplementary Table 1.

For the SLFN11$^{wt}$ with tRNA mixture dataset (Supplementary Fig. 8), particles were initially picked on 320 micrographs using Blob picker. Reasonable 2D classes were selected and used as input for Topaz train[38,39]. The resulting Topaz model was used as template for particle picking on 8,569 micrographs yielding 2,756,604 particles extracted with a box size of 320 px and a pixel size of 1.046 Å. The particles were subject to 2D classification, ab-initio reconstruction, and heterogenous refinement and the class with clearly defined features was selected (1,603,620 particles). The obtained particles were further sorted by 2D classification and heterogenous refinement resulting in five classes. The class that showed the most defined features of SLFN11 alone was selected (522,494 particles) and used for further refinement. The final resolution of the SLFN11$^{wt}$ reconstruction after non-uniform refinement[40] was 2.86 Å. From these above-mentioned five classes, two classes (249,140 particles and 126,239 particles) were analysed by 3D variability[41] for presence of an additional density belonging to bound tRNA. Resulting volumes of 3D variability served as inputs for two rounds of heterogenous refinement of the previously obtained set of

particles (1,603,620 particles). From four classes, one class (96,514 particles) showed clear features of bound tRNA to SLFN11. The final resolution of the tRNA bound SLFN11$^{wt}$ reconstruction without masking was 3.98 Å.

Cryo-EM data processing of SLFN11$^{E209A}$ dimer (Supplementary Fig. 9) was carried out in a similar fashion compared to the SLFN11$^{wt}$ with tRNA mixture dataset. For the SLFN11$^{E209A}$ data-set particles were initially picked on 2,658 micrographs using blob picker. Reasonable 2D classes were selected and used as input for Topaz train. The resulting topaz model was used as template for particle picking on 2,658 micrographs (untilted) and 4,420 micrographs (25° tilt angle), respectively. Particles corresponding to dimer classes were extracted with a box size of 320 px and a pixel size of 1.046 Å. Particles were further sorted by 2D classification, ab-initio reconstruction, and heterogenous refinement. A total of 280,205 particles were combined and subject to Ctf refinement and non-uniform refinement. The final resolution of the SLFN11$^{E209A}$ dimer reconstruction in C2 symmetry was 3.25 Å.

An analogous processing strategy was used for the SLFN11$^{E209A}$ monomer (Supplementary Fig. 9). From 3,312 micrographs, a total of 789,190 particles corresponding to monomer classes were extracted with a box size of 256 px and a pixel size of 1.046 Å. After further processing, one class with 262,713 particles was used for final 2D classification and non-uniform refinement. Processing resulted in a 4.0 Å map of the SLFN11$^{E209A}$ monomer containing 223,491 particles.

Cryo-EM data processing of SLFN11$^{wt}$ bound to ssDNA (Supplementary Fig. 10) was carried out in a similar fashion as in the SLFN11$^{E209A}$ dimer dataset. 2,331 micrographs were used for initial particle picking. The resulting Topaz model was used as template for particle picking on 2,331 micrographs and 4,088 micrographs (25° tilt angle), respectively. A total of 152,738 particles (untilted) and 310,225 particles (tilted) was extracted and combined. Further processing implying heterogenous and non-uniformed refinement with C2 symmetry applied, resulted in a 3.16 Å map of the SLFN11$^{wt}$ dimer bound to ssDNA (247,654 particles).

## Model building and refinement

Atomic models were built by rigid body docking of models predicted by AlphaFold2[42] into the cryo-EM density. For building of ssDNA, ssDNA bound to DNA2 (PDB code 5EAX (DNA2 in complex with ssDNA)) was used as a starting model. As the sequence of the bound ssDNA cannot be determined from the map, the five nucleotides closest to the 5′ end of the 60 nt sequence were modelled into the C2 symmetric ssDNA bound SLFN11 map. The models were partially rebuilt in Coot 0.9-pre[43]. Missing parts were built de-novo. Atomic models were improved by ISOLDE 1.2.2[44] and real space refinement in PHENIX 1.17[45,46] using the maps with highest resolution, respectively. The model of SLFN11$^{wt}$ bound tRNA was generated by docking of tRNA$^{Phe}$ (PDB code 5AXM] (Thg1 like protein (TLP) with tRNA$^{Phe}$)) into the corresponding density map using UCSF ChimeraX 1.2[47]. All structure figures were prepared with UCSF ChimeraX[47].

## Mass photometry

The molecular mass of SLFN11 in solution was determined by mass photometry. All mass photometry measurements were carried out using a OneMP mass photometer (Refeyn). Prior to each measurement the focus was adjusted by applying 19 µl mass photometry buffer (25 mM Tris pH 7.5, 2 mM $MgCl_2$, 1 mM DTT with variable concentrations of NaCl) to a new flow chamber. SLFN11 was diluted in sterile filtered mass photometry buffer to a final concentration of 50 nM, immediately prior to mass photometry measurements. For ssDNA stabilization experiment, 60 nt ssDNA was added with a final concentration of 100 nM or 300 nM. Movies were recorded for 60 s and data were analysed using AcquireMP (Refeyn) 2.3.

## Nuclease assay

Nuclease activity of SLFN11 was examined by a gel-based nuclease assay. The nuclease reaction was performed in nuclease buffer (25 mM Tris pH 7.5, 120 mM NaCl, 2 mM $MgCl_2$, 1 mM DTT) with 50 nM SLFN11 and 50 nM 6-FAM labelled nucleic acid substrate, unless otherwise indicated. 2 mM $MnCl_2$ was added if not stated otherwise. Reactions were started by adding the substrate and incubated at 37 °C for 45 min. Samples were mixed with loading dye (15% Ficoll, 20 mM Tris pH 7.6, 40 mM NaCl) and applied to a self-cast 15% denaturing polyacrylamide gel (Rotiphorese® DNA sequencing system). Gels were run in 0.5× TBE at 270 V (Bio-Rad) for 50 min. Gels were imaged by a Typhoon™ FLA 7000 (GE Healthcare) and analysed using GIMP 2.10.2 and ImageJ 1.8.0[48].

The effect of dimerization on the nuclease activity of SLFN11 was examined using a gel-based nuclease assay with settings described above. 30, 150, and 300 nM SLFN11[E214A] was titrated to 30 nM SLFN11[wt]. Reactions were started by adding 50 nM 6-FAM labelled tRNA[Ser].

The effect of ssDNA on the nuclease activity of SLFN11 was examined using a gel-based nuclease assay as described before. 50 nM SLFN11[wt] or SLFN11[K652D] were incubated with or without 100 nM 50 nt ssDNA. Reactions were started by adding 50 nM 6-FAM labelled tRNA[Ser].

Uncropped gels are provided in the Source Data file.

## Electrophoretic mobility shift assay (EMSA)

Binding of SLFN11 to nucleic acid substrates was monitored by electrophoretic mobility shift assay (EMSA). Either 37.5–300 nM SLFN11 (for DNA substrates) or 62.5–500 nM SLFN11 (for tRNA substrates) was incubated with 40 nM 6-FAM labelled substrates at 4 °C for 30 min in EMSA buffer (25 mM HEPES pH 7.5, 60 mM KCl, 8% glycerol, 2 mM $MgCl_2$, 1 mM DTT). Samples were mixed with loading dye (15% Ficoll, 20 mM Tris pH 7.6, 40 mM NaCl) and applied to a NativePAGE 3–12% Bis-Tris gel (Thermo Fisher). The electrophoresis was performed in 1× NativePAGE running buffer (Thermo Fisher) at 100 V for 120 min at 4 °C. The gels were imaged using a Typhoon™ FLA 7000 (GE Healthcare) and analysed using GIMP. Uncropped gels are provided in the Source Data file.

## Affinity measurement by fluorescence anisotropy

Initial protein dilutions (0, 6.25, 12.5, 25, 50, 100, 200, 400, and 800 nM) of SLFN11 were prepared in assay buffer (25 mM Tris pH 7.5, 120 mM NaCl, 2 mM $MgCl_2$, 1 mM DTT). Protein dilutions were then mixed with 6-FAM labelled DNA (Supplementary Table 2) in assay buffer without NaCl (final DNA concentration of 10 nM) in a 1:1 (v/v) ratio (final volume: 20 µl, Greiner Flat Bottom Black 384 well plate). The reaction was incubated at 25 °C for 30 min and the fluorescence anisotropy was subsequently measured at an excitation wavelength of 470 nm and an emission wavelength of 520 nm using an Infinite M1000 microplate photometer (Tecan). Experiments were performed at least three times. The background signal (no protein sample) was subtracted from each value of a dilution series and the datasets were analysed with Prism 6.07 (GraphPad Software). The datasets were fit to a Hill model to determine the apparent dissociation constants.

## Nano differential scanning fluorimetry (nanoDSF)

Binding of SLFN11 to various substrates was examined by nanoDSF (Tycho NT.6, NanoTemper Technologies). 300 nM SLFN11 was incubated with 300 nM of 50 nt ssDNA or 50 bp dsDNA in nanoDSF buffer (25 mM Tris pH 7.5, 60 mM NaCl, 2 mM $MgCl_2$, 1 mM DTT) for 30 min on ice, respectively. For testing of dependence on salt (NaCl) and ions ($Mg^{2+}$, $Mn^{2+}$, $Ca^{2+}$, and $Zn^{2+}$), nanoDSF buffer was adjusted accordingly. Interaction with nucleotides was performed similarly, where SLFN11 or SLFN5 (300 nM) were incubated with or without corresponding nucleotides (1 mM) in nanoDSF buffer. The samples were loaded into glass capillaries and the internal fluorescence at 330 nm and 350 nm was measured while a thermal gradient was applied. Data were analysed using the internal Tycho NT.6 software 1.3.2.878 and plotted with Prism (GraphPad Software).

## ATP hydrolysis assay

A fluorescence-based ATPase assay was conducted to determine the ATPase rate of SLFN11[49]. SLFN11 (250 nM) was incubated with 150 nM of different DNA or RNA substrates in ATPase buffer (25 mM Tris pH 7.5, 50 mM NaCl, 1 mM DTT, 2 mM $MgCl_2$, 0.1 mg ml$^{-1}$ BSA) at 4 °C for 30 min. RPA was used at a concentration of 250 nM. SLFN11:substrate complexes were combined with 0.1 mM NADH in reaction buffer (25 mM Tris pH 7.5, 50 mM NaCl, 1 mM DTT, 2 mM $MgCl_2$, 0.1 mg ml$^{-1}$ BSA, 0.5 mM PEP (phosphoenolpyruvate) , 1 mM ATP, 25 U ml$^{-1}$ lactate dehydrogenase/ pyruvate kinase (Sigma-Aldrich)) in a 384 well plate (Greiner). Hexokinase from *Saccharomyces cerevisiae* (1.5 nM, Sigma-Aldrich) supplemented with 300 µM glucose served as the positive control. The fluorescence of NADH was measured at 37 °C using an Infinite M1000 microplate photometer (Tecan). The reaction was monitored for 45 min (20 s intervals) using an excitation wavelength of 340 nm and an emission wavelength of 460 nm. Data were analysed using Prism (GraphPad Software).

## tRNA sequencing

For the identification of the SLFN11 cleavage position in tRNA[Ser], a nuclease reaction was performed as described above. Briefly, 1 µM tRNA substrate in nuclease buffer (25 mM Tris pH 7.5, 120 mM NaCl, 2 mM $MgCl_2$, 1 mM DTT, 2 mM $MnCl_2$) was incubated with 1.1 µM SLFN11 at 37 °C for 45 min. 15 µl of the reaction mixture were purified on a Sephadex G-25 column (Roche Quick Spin columns for radiolabeled RNA). The eluate (20 ng µl$^{-1}$) was converted to an Illumina sequencing library with the SMARTer smRNA-Seq kit (Takara) following the manufacturer's instruction. The RNA was enzymatically polyadenylated, then reversely transcribed with an oligo dT primer and a template-switching primer, obtaining a cDNA that was extended with primer-templated sequences on both ends of the original RNA. The cDNA was amplified with barcoded primers, converting it into a sequencing-ready Illumina-compatible library. The cDNA library was sequenced on a NextSeq1000 (Thermo Fischer) in 60 bp paired-end mode. The obtained sequencing reads were demultiplexed and poly-A tails at the end as well as three nucleotides at the beginning, that were introduced by the template-switching mechanism, were removed. The reads were mapped to the sequence of tRNA[Ser] with Burrows-Wheeler Aligner 0.17.7[50]. The start positions of the mapped reads were visualized in a histogram using Prism (GraphPad Software).

## Mass spectrometry

2 µl of purified SLFN11 (3.2 µM, purified from *Trichoplusia ni* High Five cells) were diluted with 5 µl 100 mM $NH_4HCO_3$ and 3 µl water. Proteins were reduced by addition of 1 µl DTE (50 mM in 50 mM $NH_4HCO_3$) and incubated for 30 min at 37 °C. Carbamidomethylation of cysteines was performed by adding 2 µl of iodoacetamide (100 mM in 50 mM $NH_4HCO_3$) and 30 min incubation at RT. For protein digestion, 20 ng of Trypsin (Promega) was added to the sample and incubated at 37 °C overnight. After digestion, the sample was acidified with 2 µl 15% formic acid. Liquid chromatography-mass spectrometry analysis was performed on an Ultimate 3000 nano-LC system (Thermo Fisher) coupled with a Q Exactive HF-X mass spectrometer (Thermo Fisher). Peptides were separated with an EasySpray reversed-phase column (PepMap RSLC C18, 50 cm length, 75 µm ID, Thermo Fisher) at a flow rate of 250 nl min$^{-1}$. Solvent A consisted of 0.1% formic acid in water and solvent B of 0.1% formic acid in acetonitrile. The chromatography method included gradients from 3% to 25% solvent B in 30 min and from 25% to 40% B in 5 min. For data acquisition, a top 12 data-dependent acquisition method was used. Spectra were searched using MASCOT 2.4[51] (Matrix Science Ltd) and the human subset of the Swiss-Prot database[52].

## Reporting summary

Further information on research design is available in the Nature Research Reporting Summary linked to this article.

## Data availability

The data that support this study are available from the corresponding author upon request. The coordinates of the SLFN11$^{wt}$, SLFN11$^{E209A}$ dimer, and SLFN11$^{wt}$ ssDNA-bound structure have been deposited in the Protein Data Bank (PDB) under the accession code 7ZEL, 7ZEP, and 7ZES, respectively. The SLFN11$^{wt}$ dimer cryo-EM reconstruction is available at the Electron Microscopy Data Bank (EMDB) under the EMBD accession code EMD-14690. The SLFN11 dimer reconstruction of SLFN11$^{wt}$ bound to tRNA is available at the EMDB under the EMDB accession code EMD-14695. The SLFN11 monomer and dimer reconstruction of SLFN11$^{E209A}$ is available at the EMDB under the EMDB accession code EMD-14693 and EMD-14691, respectively. The SLFN11 dimer reconstruction of SLFN11$^{wt}$ bound to ssDNA is available at the EMDB under the accession code EMD-14692. MS spectra were searched using the human subset of the Swiss-Prot database [https://www.uniprot.org/]. Source data are provided with this paper.

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

## Acknowledgements

We acknowledge Dr. Thomas Fröhlich from the Gene Center at the LMU for mass spectrometry analysis and data evaluation. We acknowledge Dr. Daniel Bollschweiler from the Max-Planck Institute of Biochemistry for the mass photometry experiments and data evaluation. We thank all members of the Hopfner lab for helpful discussions. We acknowledge support by the Deutsche Forschungsgemeinschaft (DFG, German Research Foundation) – Project-ID 210592381 – SFB 1054 (to K.L.), the Gottfried Wilhelm Leibniz-Prize (to K.-P.H.) and the European Research Council (ERC Advanced Grant INO3D, to K.-P.H.).

## Author contributions

F.J.M., S.J.W. and K.L. conceived the project. F.J.M., S.J.W., M.K. and K.L. designed all structural and biochemical experiments. F.J.M., S.J.W. and M.K., conducted all structural and biochemical experiments. F.J.M. S.J.W., M.K. and K.L. carried out the cryo-EM data collection and analysis. S.K. performed tRNA sequencing analysis. K.-P.H. helped with the analysis and interpretation of the results. F.J.M., S.J.W., M.K. and K.L. wrote the manuscript. K.L. and K.-P.H. provided funding. All authors discussed and commented on the results and the manuscript.

## Funding

## Competing interests

The authors declare no competing interests.
