## [Peer Review File · Nature Communications]

Mechanistic understanding of human SLFN11REVIEWER COMMENTS

Reviewer #1 (Remarks to the Author):

Outstanding structural and mechanistic analysis of Slfn binding to various nucleic acids, and elucidation of its regulation. Experiments are well designed and controlled, as such, there is little criticism on the data presented.

There are some minor issues that should be addressed nevertheless:

*) Is there any evidence that Slfns can form heterodimers

*) Fig 3a: The authors argue that Slfn11 only binds ssDNA, but not dsDNA. However, the possibility is not addressed that binding to dsDNA might simply require different conditions (e.g salt conc, ion composition etc). Does the sequence of the dsDNA matter (Palindromic?)

*) Several inhibitory phosphorylation sites have been identified in Slfn11 (ref 33), but there is little analysis or discussion of the impact of the phosphorylation/dephosphorylation of these residues on tRNA or ss/dsDNA binding.

*) Relating to the above question: is the Slfn11 used in these studies phosphorylated on the residues identified in Ref 33? Can S->A mutants of these sites cause dsDNA binding?

Reviewer #2 (Remarks to the Author):

Two apparently distinct functions of SLFN11 have been described in the literature. First, the SLFN11 RNase cleaves type II tRNAs, downregulating translation of viral proteins and specific DNA damage response proteins with elevated frequency of Leu-TTA codons. In addition, SLFN11 is also recruited to sites of DNA damage and stalled replication forks, blocking fork progression and sensitizing cells to DNA damaging agents. In this manuscript, the authors take a structural biology approach to understanding the regulation of these two activities, using Cryo-EM to solve the structure of full-length human SLFN11 apoenzyme, SLFN11 bound to yeast tRNA, and SLFN11 bound to single-stranded DNA.

The authors found that SLFN11 forms a salt-sensitive dimer anchored by two interaction interfaces. Dimerization stabilizes the C-lobes of the SLFN11 core domains, which are flexible in the monomer. The SLFN11 dimer bound a single tRNA, and substrate recognition involved both N and C lobes of both protomers of the dimer. SLFN11 bound both tRNA-Met (type I) and tRNA-Ser (type II) but only cleaved tRNA-Ser near the base of the acceptor stem. Titration of an RNase-inactive mutant of SLFN11 into a mix with wild-type increased nuclease activity, indicating that dimerization potentiates SLFN11 RNase activity.

The authors also report that the SLFN11 helicase domain binds single-stranded but not double-stranded DNA, and addition of single-stranded DNA shifts SLFN11 equilibrium to the dimer state. Because each helicase domain of a dimer can bind single stranded DNA separately, the authors suggest that the SLFN11 dimer can bind to both strands of single stranded DNA at a stalled replication fork and block fork progression. However, the authors found that although SLFN5 can bind ATP and ATP γ S, SLFN11 cannot due to steric hindrance. The ATPase domains of the SLFN11 dimer are thus locked in an inactive state. SLFN11 is also missing a “pin motif” used for double stranded unwinding by some helicases, also suggesting the SLFN11 is not a functional helicase.

The experiments described in this manuscript are well done, resulting in major advances in understanding of the biochemistry and function of SLFN11. I have no major concerns about the quality of the work presented in this manuscript. However, I do have some questions and points to consider:

1. It should be possible to sequence the tRNA fragments to precisely identify the cleavage site, rather than approximate.
2. The authors state that “the ribonuclease activity of subgroup II Schlafen family member SLFN12 is stimulated by dimerization.” However, this is still only a hypothesis and remains to be experimentally proven.
3. Extended Data Figure 5a requires more explanation to permit interpretation by readers who are not familiar with this technique.
4. If K652 is required for ssDNA binding, and ssDNA binding promotes the dimerized state, and the SLFN11 dimer is the RNase active state, why doesn't K652D decrease SLFN11 RNase activity?
5. Is the helicase pin motif found in SLFN5 or SLFN13? Are these two proteins thought to be helicase-active?

6. None of the SLFN11 homodimerization interfaces were probed or validated by site-directed mutagenesis. The authors should either perform these experiments or explain why they would not be informative.

Reviewer #3 (Remarks to the Author):

Metzner et al. present a thorough functional and structural characterization of the human SLFN11 protein. SLFN11 is a bifunctional enzyme possessing an ATPase and nuclease activity. It is known that SLFN11 can sensitize tumor cells to DNA damaging agents and counteract viral infections such as HIV-1. However, it remains poorly understood by which mechanisms SLFN11 achieves these two functions. Metzner et al. first introduce the apo structures of SLFN11 in dimeric and monomeric forms determined by single-particle cryo-EM. Additionally, they solved the cryo-EM structure of SLFN11 bound to a tRNA substrate and show that SLFN11 binds the tRNA between two lobes of its core domain. They further characterize binding of different tRNA substrates to SLFN11 and show that SLFN11 is catalytically active and cleaves specific tRNA substrates. Mutational mapping reveals that amino acids 209, 214 and 216 of the nuclease active site are crucial for nuclease activity. The authors also characterize the second mode of action of SLFN11, which is DNA binding and an ATPase activity. They demonstrate specificity of SLFN11 for binding ssDNA and determine the binding sites by solving the ssDNA-SLFN11 complex structure by single particle cryo-EM. However, SLFN11 remained inactive in ATPase assays, which the authors find to be caused by an autoinhibitory conformation of SLFN11. The authors summarize that their study gives some insights in how SLFN11 is regulated and how it binds to replication forks. They further conclude, that the structural and biochemical framework provided in this study lays the basis for future in cellulo investigations of SLFN11 function.

Although the authors discuss the structure and function of SLFN11 in great depth, the functional implications of tRNase activity and ssDNA binding in the context of stalled replication forks and antiviral immunity should be discussed in more detail (see major points). The experimental work is done well and only requires minor clarifications.

After these issues are addressed by the authors, I see no reason to not recommend the manuscript for publication.

Major points:

1) The title of the submitted manuscript is “Mechanistic understanding of human SLFN11 in antiviral immunity and cancer”, however, the study in its current form does not provide much insights in the two processes. It rather gives a detailed characterization of substrate binding and enzymatic activities. The authors should extend the discussion to relate their findings more broadly with antiviral immunity and cancer biology. If the findings of the study are too limited to provide a clear “understanding of human SLFN11 in antiviral immunity and cancer”, the title should be revised (like “Mechanistic understanding of human SLFN11 tRNase and ATPase activities”).

2) The authors claim that they presented insight into the regulation of SLFN11 and how it might bind to replication forks (lines 267+). Except for mentioning that SLFN11 alone cannot function as a strand opening helicase, the authors do not discuss this function any further. Do the authors think that SLFN11 could bind dsDNA after an activation step? Or is the ssDNA the substrate of a strand opening helicase activity (if it is one)? What could be the activation step that is necessary to switch the ATPase on? The manuscript would greatly benefit from a more in-depth discussion here, also on the context of the function of other (related) Schlafen proteins.

Minor points:

1) It might be beneficial to mention early in the summary and introduction that SLFN11 has two enzymatic activities and what they are. Otherwise, a reader who is not aware of this might miss, how the two cellular functions arise.

2) The introduction does not mention other structural information about Schlafen family proteins (there is at least an x-ray structure of SLFN5). Since the study is extremely focused on structure and function of a Schlafen protein, it is important to introduce the structural work that has been done (and why it is necessary to solve the structure of SLFN11).

3) Line 57 (and throughout): The authors use “Slfn” as an abbreviation for Schlafen, however, in all figures it is written in the long form. Maybe the authors can decide to stick to one form for text and figures.

4) Overall structure of the dimeric apoenzyme: The authors describe the dimer and monomer structures as if they were solved from a single data set (lines 61+). However, the reported resolution of the dimer (2.8Å) does not fit with ED Fig. 7 (reporting 3.2Å). Was the dimer structure derived from the wt data set (ED Fig 6, reporting 2.8Å)? Further, the referenced ED Fig 1 legend reports the dimer resolution as 3.2 Å, whereas the text reports 2.8 Å. The authors should write more clearly which reconstructions they used to derive at the dimer and monomer models.

5) The N/C-lobe nomenclature in the text is somewhat confusing as it is used for the nuclease as well as the ATPase parts. Especially that the text often just mentions N or C-lobes without referencing to the respective ATPase/nuclease part of SLFN11. The authors should make this clearer in the text.

6) In ED Fig 3b, the authors show binding of two different tRNAs to wt SLFN11. In the text at line 116 the authors state that SLFN11 binds both substrates with “comparable affinity”. Although EMSAs are not the most accurate assays for quantification of affinities, there is an obvious difference between the two substrates (tRNA-Met is bound better than tRNA-Ser). This could reflect the specificity of SLFN11 towards tRNA-Ser and should be mentioned in the text.

7) The authors refined the ssDNA-SLFN11 complex reconstruction using C2 symmetry. However, it does not seem clear that the bound ssDNA has true C2 symmetry. The ssDNA substrate that was added for the complex formation was a 60 nt long and any part of it could be bound. Could the authors explain which reconstruction they used for model building of the ssDNA and how the C1 and C2 reconstruction compare? Can the authors speculate on any sequence specificity of SLFN11?

8) The method section also does not contain information on how the ssDNA was modeled. Especially what sequence (see also point above). The authors should add this along with an ED Fig panel showing the density for ssDNA (maybe C1 and C2 symmetry side by side).

9) The cryo-EM data collection methods do not contain the number of frames taken per movie. Please add this.

10) ED Fig 6 does not state that C2 symmetry was imposed during refinements. Please add that information.

11) The angular distribution plots (and the 25 degree tilt approach) indicate problems with preferred particle orientation. Could the authors provide 3D FSC plots to confirm the isotropy of the resolution they report?

Signed review: Christian Dienemann

We thank the editor and all reviewers for their positive feedback as well as their constructive and insightful comments. We addressed all reviewers' points and revised the manuscript along their suggestions.

In summary we added the following points:

In order to analyse the phosphorylation status of the SLFN11 protein expressed from insect cell we used mass spectrometry analysis. We analysed the phospho-mimetic mutant S753D concerning its ssDNA binding ability, which added significant information in terms of SLFN11's regulation. Further, we proved that the endonucleolytic cleavage position of SLFN11 is exactly 10 bases from the 3' end of the tRNA by additional sequencing data. We also included dsDNA binding assays under various buffer conditions which proved that SLFN11 cannot bind dsDNA on its own.

The reviewers also asked for a more elaborate and speculative discussion of our data in light of the existing literature. Initially we had to shorten our manuscript substantially to meet the terms of the journal. Now we included a more comprehensive discussion part in the revised version of the manuscript.

All in all, we believe that our manuscript is substantially improved and we would like to thank the reviewers for their input which led to insightful results.

Point-by-point response to referees:

Reviewer #1 (Remarks to the Author):

Outstanding structural and mechanistic analysis of Slfn binding to various nucleic acids, and elucidation of its regulation. Experiments are well designed and controlled, as such, there is little criticism on the data presented.

We thank the reviewer for the kind appreciation of our work and we hope that we address all issues in the revised version of the manuscript.

There are some minor issues that should be addressed nevertheless:

*) Is there any evidence that Slfns can form heterodimers

We are not aware of any evidence in the literature of Slfn proteins forming heterodimers. Homodimerization has been shown for SLFN11 (this study) and SLFN12 (Garvie et al., 2021; Chen et al., 2021). In our previous study on SLFN5 (Metzner et al., 2021), we performed cryo-EM analysis of full length SLFN5 resulting only in monomeric SLFN5 structures. A study on the rat Slfn13 Slfn core domain found monomeric rSlfn13¹⁻³⁵³ and dimeric His6-tagged rSlfn13¹⁴⁻³⁵³. In this case, dimerization was suggested to be an artefact of the His₆-tag.

Residues of the dimerization interface are somewhat conserved between SLFN11 and SLFN13 and less conserved between SLFN11 and SLFN5, SLFN12 or SLFN14. However, Slfn proteins are functionally divergent and differentially expressed in various tissues and species. Thus, we speculate that solely homodimerization is a common activation step of the nucleolytically active Slfn proteins.

*) Fig 3a: The authors argue that Slfn11 only binds ssDNA, but not dsDNA. However, the possibility is not addressed that binding to dsDNA might simply require different conditions (e.g salt conc, ion composition etc). Does the sequence of the dsDNA matter (Palindromic?)

We now provide nanoDSF based dsDNA binding measurements using a wide range of buffer conditions (20 mM NaCl – 300 mM NaCl, Supplementary Fig. 5g), different ions (Mg^{2+} , Mn^{2+} , Zn^{2+} and Ca^{2+} , Supplementary Fig. 5h) and various dsDNA sequences (Supplementary Fig. 5f). We could not detect any dsDNA binding. The ssDNA bound structure also implies that at the position where ssDNA is bound a regular B-form dsDNA would not fit based on the shape of the binding site (Supplementary Fig. 5e). Finally, as illustrated in Supplementary Fig. 7b, the common dsDNA binding domains (1B and 2B) as well as the characteristic strand separating Pin motif of other SF1 helicases are missing in SLFN11. Thus, we conclude that SLFN11 is not able to bind dsDNA on its own.

*) Several inhibitory phosphorylation sites have been identified in Slfn11 (ref 33), but there is little analysis or discussion of the impact of the phosphorylation/dephosphorylation of these residues on tRNA or ss/dsDNA binding.

*) Relating to the above question: is the Slfn11 used in these studies phosphorylated on the residues identified in Ref 33? Can S->A mutants of these sites cause dsDNA binding?

Malone et al. reported that dephosphorylation is necessary to functionally activate SLFN11. Thereby substitution of the amino acids S219, T230 and S753 with Ala retained SLFN11 function, whereas replacement with the phosphomimic aspartate rendered the resulting proteins inactive. Therefore, we examined the phosphorylation state of the SLFN11 protein expressed from insect cells used for our studies via mass spectrometry. The data show that all three sites S219, T230 and S753 are unphosphorylated (please see new Supplementary Fig. 4). This is in accordance with the observed tRNA cleavage activity of the SLFN11 protein used. Based on the tRNA bound cryo-EM structure we suggest that phosphorylation of residues S219 and T230 directly interferes with tRNA binding (see Supplementary Fig. 4a). We added the mass spectrometry data in the result part of the paper (line 178).

Since residue S753 is not phosphorylated in the protein used in our studies we think it is not necessary to mutate it to alanine. However, the influence of S753 phosphorylation on SLFN11 function is less obvious from the structural point of view. Thus, we mutated S753 to the phosphomimic aspartate and tested for ssDNA and dsDNA binding activity. The S753D mutant is not only deficient in tRNA cleavage as reported by Malone et al. but EMSAs indicate that it is also unable to bind ss/dsDNA (Supplementary Fig. 6).

Reviewer #2 (Remarks to the Author):

Two apparently distinct functions of SLFN11 have been described in the literature. First, the SLFN11 RNase cleaves type II tRNAs, downregulating translation of viral proteins and specific DNA damage response proteins with elevated frequency of Leu-TTA codons. In addition, SLFN11 is also recruited to sites of DNA damage and stalled replication forks, blocking fork progression and sensitizing cells to DNA damaging agents. In this manuscript, the authors take a structural biology approach to understanding the regulation of these two activities, using Cryo-EM to solve the structure of full-length human SLFN11 apoenzyme, SLFN11 bound to yeast tRNA, and SLFN11 bound to single-stranded DNA.

The authors found that SLFN11 forms a salt-sensitive dimer anchored by two interaction interfaces. Dimerization stabilizes the C-lobes of the SLFN11 core domains, which are flexible in the monomer. The SLFN11 dimer bound a single tRNA, and substrate recognition involved both N and C lobes of both protomers of the dimer. SLFN11 bound both tRNA-Met (type I) and tRNA-Ser (type II) but only cleaved tRNA-Ser near the base of the acceptor stem. Titration of an RNase-inactive mutant of SLFN11 into a mix with wild-type increased nuclease activity, indicating that dimerization potentiates SLFN11 RNase activity.

The authors also report that the SLFN11 helicase domain binds single-stranded but not double-stranded DNA, and addition of single-stranded DNA shifts SLFN11 equilibrium to the dimer state. Because each helicase domain of a dimer can bind single stranded DNA separately, the authors suggest that the SLFN11 dimer can bind to both strands of single stranded DNA at a stalled replication fork and block fork progression. However, the authors found that although SLFN5 can bind ATP and ATP γ S, SLFN11 cannot due to steric hindrance. The ATPase domains of the SLFN11 dimer are thus locked in an inactive state. SLFN11 is also missing a “pin motif” used for double stranded unwinding by some helicases, also suggesting the SLFN11 is not a functional helicase.

The experiments described in this manuscript are well done, resulting in major advances in understanding of the biochemistry and function of SLFN11. I have no major concerns about the quality of the work presented in this manuscript. However, I do have some questions and points to consider:

We thank the reviewer for his insightful comments and hope to answer all questions raised with the revised version of the manuscript.

1. It should be possible to sequence the tRNA fragments to precisely identify the cleavage site, rather than approximate.

As suggested, we analysed the position of the endonucleolytic cleavage site in type II tRNA^{Ser} by sequencing experiments. We added the corresponding data to the manuscript (line 135, method section and Supplementary Fig. 3e) which prove that SLFN11 cleaves exactly 10 bases from the 3' end of tRNA^{Ser}. This is in accordance with the estimated position of the tRNA in the cryo-EM structure (Supplementary Fig. 3d) and the cleavage product identified in the nuclease assay. The result hints towards a single active site in the SLFN11 dimer, despite the fact that the tRNA is in close proximity to both catalytic centres in the complex structure.

2. The authors state that “the ribonuclease activity of subgroup II Schlafen family member SLFN12 is stimulated by dimerization.” However, this is still only a hypothesis and remains to be experimentally proven.

We changed the sentence in the manuscript to: "...it was hypothesized..."

3. Extended Data Figure 5a requires more explanation to permit interpretation by readers who are not familiar with this technique.

We explain the assay in more detail in the figure caption (Supplementary Fig. 7a) and method section of the manuscript to facilitate interpretation of the results by the reader. Briefly, the hydrolysis of ATP to ADP is linked to the oxidation of NADH to NAD⁺ by the combined action of pyruvate kinase and L-lactate dehydrogenase. The fluorescence of NADH is measured over time. ATP consumption is enzymatically coupled to the oxidation of NADH, causing a decrease in fluorescence upon ATP hydrolysis.

4. If K652 is required for ssDNA binding, and ssDNA binding promotes the dimerized state, and the SLFN11 dimer is the RNase active state, why doesn't K652D decrease SLFN11 RNase activity?

In general, the nuclease assays were performed without ssDNA present. However, in Supplementary Fig. 5d we specifically tested the influence of ssDNA on RNase activity and observed a slight stimulation of SLFN11^{wt} nuclease activity in presence of ssDNA. We cannot recognize this minor stimulation with the K652D mutant (Supplementary Fig. 5d). Since the effect is not significant, we propose that the ssDNA has no direct effect on the ribonucleolytic activity and it is rather the stabilization of the dimeric SLFN11 form by ssDNA which induces the slight enhancement.

5. Is the helicase pin motif found in SLFN5 or SLFN13? Are these two proteins thought to be helicase-active?

The Pin motif as well as the dsDNA binding domains 1B and 2B of other helicases with strand-opening activity cannot be found in SLFN5 or SLFN13 or other Slfn family members (Supplementary Fig. 7b). Hence, we propose that the Slfn proteins are no strand-opening helicases on their own. A so far unidentified binding partner might be required for helicase activity or the ATPase domain could have another function such as binding to or translating along ssDNA.

6. None of the SLFN11 homodimerization interfaces were probed or validated by site-directed mutagenesis. The authors should either perform these experiments or explain why they would not be informative.

This experiment is shown in Supplementary Fig. 3g and was unfortunately overlooked. The Interface I mutant has no effect on ribonuclease activity, while ribonuclease activity is abolished in the Interface I + II double mutant.

Reviewer #3 (Remarks to the Author):

Metzner et al. present a thorough functional and structural characterization of the human SLFN11 protein. SLFN11 is a bifunctional enzyme possessing an ATPase and nuclease activity. It is known that SLFN11 can sensitize tumor cells to DNA damaging agents and counteract viral infections such as HIV-1. However, it remains poorly understood by which mechanisms SLFN11 achieves these two functions. Metzner et al. first introduce the apo structures of SLFN11 in dimeric and monomeric forms determined by single-particle cryo-EM. Additionally, they solved the cryo-EM structure of SLFN11 bound to a tRNA substrate and show that SLFN11 binds the tRNA between two lobes of its core domain. They further characterize binding of different tRNA substrates to SLFN11 and show that SLFN11 is catalytically active and cleaves specific tRNA substrates. Mutational mapping reveals that amino acids 209, 214 and 216 of the nuclease active site are crucial for nuclease activity. The authors also characterize the second mode of action of SLFN11, which is DNA binding and an ATPase activity. They demonstrate specificity of SLFN11 for binding ssDNA and determine the binding sites by solving the ssDNA-SLFN11 complex structure by single particle cryo-EM. However, SLFN11 remained inactive in ATPase assays, which the authors find to be caused by an autoinhibitory conformation of SLFN11. The authors summarize that their study gives some insights in how SLFN11 is regulated and how it binds to replication forks. They further conclude, that the structural and biochemical framework provided in this study lays the basis for future in cellulo investigations of SLFN11 function. Although the authors discuss the structure and function of SLFN11 in great depth, the functional implications of tRNase activity and ssDNA binding in the context of stalled replication forks and antiviral immunity should be discussed in more detail (see major points). The experimental work is done well and only requires minor clarifications. After these issues are addressed by the authors, I see no reason to not recommend the manuscript for publication.

We thank the reviewer for his insightful comments and hope to answer all questions raised with the revised version of the manuscript. We apologize for the rather brief discussion of our data in the initial version, but in order to include all data and meet the word limitations of the journal we had to shorten the manuscript.

Major points:

1) The title of the submitted manuscript is “Mechanistic understanding of human SLFN11 in antiviral immunity and cancer”, however, the study in its current form does not provide much insights in the two processes. It rather gives a detailed characterization of substrate binding and enzymatic activities. The authors should extend the discussion to relate their findings more broadly with antiviral immunity and cancer biology. If the findings of the study are too limited to provide a clear “understanding of human SLFN11 in antiviral immunity and cancer”, the title should be revised (like “Mechanistic understanding of human SLFN11 tRNase and ATPase activities”).

We changed the title to: “Mechanistic understanding of human SLFN11”

2) The authors claim that they presented insight into the regulation of SLFN11 and how it might bind to replication forks (lines 267+). Except for mentioning that SLFN11 alone cannot function as a strand opening helicase, the authors do not discuss this function any further. Do the authors think that SLFN11 could bind dsDNA after an activation step? Or is the ssDNA the substrate of a strand opening helicase activity (if it is one)? What could be the activation step that is necessary to switch the ATPase on? The manuscript would greatly benefit from a more in-depth discussion here, also on the context of the function of other (related) Schlafen proteins.

In the revised version of the manuscript, we include a more in-depth discussion on the SLFN11 ATPase/helicase activity.

In SLFN11 as well as in other subgroup III Slfn family members, the DNA strand opening Pin motif as well as the dsDNA binding domains 1B and 2B, that are often found in helicases with strand-opening activity, cannot be found. A structural comparison to other helicases is illustrated in Supplementary Fig. 7b. In contrast to SLFN11, human SLFN5 can bind ATP and dsDNA *in vitro*. However, ATP hydrolysis or helicase activity was not detected for SLFN5 (Metzner et al., 2021). Hence, we propose that the Slfn proteins are no strand opening helicases on their own. Maybe a so far unknown binding partner or modification is required for helicase activity or the ATPase domain might have another function, such as binding to or translating along ssDNA. Nevertheless, an analysis of the SLFN11 Walker B E669Q mutant (Murai et al., 2018) showed that the conserved Walker B motif of the ATPase domain is necessary for drug-induced cell killing and replication blockage. However, we could not show ATP binding nor hydrolysis by SLFN11 *in vitro* (Fig. 4a, Supplementary Fig. 7a). This discrepancy is unfortunately still unresolved and needs to be answered in future studies. We now include a more elaborate dsDNA binding analysis using different buffers, ions and dsDNA substrates (Supplementary Fig. 5f-h) but could not identify conditions that facilitate dsDNA binding.

Minor points:

1) It might be beneficial to mention early in the summary and introduction that SLFN11 has two enzymatic activities and what they are. Otherwise, a reader who is not aware of this might miss, how the two cellular functions arise.

We mention the endonuclease and replication fork binding activities in the abstract now.

2) The introduction does not mention other structural information about Schlafen family proteins (there is at least an x-ray structure of SLFN5). Since the study is extremely focused on structure and function of a Schlafen protein, it is important to introduce the structural work that has been done (and why it is necessary to solve the structure of SLFN11).

We now mention all crystal structures (SLFN5 and rSLFN13 N-terminus) and cryo-EM structures of SLFN5 and SLFN12 in the introduction. Although several structures have been solved, information about substrate recognition and processing as well as the C-terminal helicase domain is still missing. Moreover, the helicase domain of SLFN11 adopts a different orientation with respect to the linker domain when compared to SLFN5, which provides important information for all subgroup III Slfn family members. Further, there was no structural information available concerning substrate recognition or processing.

3) Line 57 (and throughout): The authors use “Slfn” as an abbreviation for Schlafen, however, in all figures it is written in the long form. Maybe the authors can decide to stick to one form for text and figures.

We replaced “Schlafen” with the abbreviation “Slfn” in the text and figures.

4) Overall structure of the dimeric apoenzyme: The authors describe the dimer and monomer structures as if they were solved from a single data set (lines 61+). However, the reported resolution of the dimer (2.8Å) does not fit with ED Fig. 7 (reporting 3.2Å). Was the dimer structure derived from the wt data set (ED Fig 6, reporting 2.8Å)? Further, the referenced ED Fig 1 legend reports the dimer resolution as 3.2 Å, whereas the text reports 2.8 Å. The authors should write more clearly which reconstructions they used to derive at the dimer and monomer models.

We edited the mentioned paragraph in the results part, to describe the different datasets and the respective reconstructions in more detail.

The SLFN11^{wt} dimer (2.86 Å), SLFN11^{E209A} monomer (4.0 Å) and SLFN11^{E209A} (3.25 Å) dimer reconstructions were obtained from three separate datasets. The SLFN11^{wt} dimer (2.86 Å) was reconstructed from the SLFN11^{wt} with tRNA dataset, using the particles that are not bound to tRNA (Supplementary Fig. 8). This reconstruction was used to model the SLFN11^{wt} dimer structure (Fig. 1b).

The SLFN11^{E209A} datasets showed monomer as well as dimer classes, however, the monomer-dimer ratios varied between the datasets. Thus, we reconstructed the monomeric (4.0 Å) and dimeric (3.25 Å) SLFN11^{E209A} maps from two different dataset to obtain good resolution (Supplementary Fig. 1a, Supplementary Fig. 9).

5) The N/C-lobe nomenclature in the text is somewhat confusing as it is used for the nuclease as well as the ATPase parts. Especially that the text often just mentions N or C-lobes without referencing to the respective ATPase/nuclease part of SLFN11. The authors should make this clearer in the text.

We always mention helicase/Slfn core when talking about N-/ C-lobe in the revised manuscript.

6) In ED Fig 3b, the authors show binding of two different tRNAs to wt SLFN11. In the text at line 116 the authors state that SLFN11 binds both substrates with “comparable affinity”. Although EMSAs are not the most accurate assays for quantification of affinities, there is an obvious difference between the two substrates (tRNA-Met is bound better than tRNA-Ser). This could reflect the specificity of SLFN11 towards tRNA-Ser and should be mentioned in the text.

We are aware of the small difference in affinity between tRNA^{Met} and tRNA^{Ser} as detected by EMSA and tried to determine the affinities by a more quantitative method. However, as anisotropy measurements with SLFN11 and tRNA did not yield reproducible results, we do not want to draw the conclusion that the nuclease specificity of SLFN11 is directly correlated with its affinity to tRNA.

7) The authors refined the ssDNA-SLFN11 complex reconstruction using C2 symmetry. However, it does not seem clear that the bound ssDNA has true C2 symmetry. The ssDNA substrate that was added for the complex formation was a 60 nt long and any part of it could be bound. Could the authors explain which reconstruction they used for model building of the ssDNA and how the C1 and C2 reconstruction compare? Can the authors speculate on any sequence specificity of SLFN11?

8) The method section also does not contain information on how the ssDNA was modeled. Especially what sequence (see also point above). The authors should add this along with an ED Fig panel showing the density for ssDNA (maybe C1 and C2 symmetry side by side).

When no symmetry is imposed and a low resolution reconstruction originating from a heterogeneous refinement job is regarded, the paths of the ssDNAs extending away from the ssDNA binding sites seem to differ between the two molecules. However, when refining to high resolution without imposition of symmetry, the reconstruction shows very similar ssDNA binding modes at both sites. We provide a comparison of the ssDNA binding regions of the C1 and C2 maps (Supplementary Fig. 10g) showing that there are no large differences in the ssDNA binding mode.

We now include information on ssDNA modelling in the methods section. Due to the higher resolution, the C2 map was used for model building and a structure of DNA2 bound to ssDNA was used as starting model (PDB code 5EAX). As the reconstruction does not allow to determine the sequence of the bound ssDNA, the first 5 nucleotides of the 60 nt sequence were modelled.

SLFN11 mostly interacts with ssDNA via the phosphate backbone. However, some interactions with the bases might indicate some degree of sequence specificity. This is supported by the observation that ssDNAs with different sequences show moderately different affinities in an EMSA (see Figure below). In a biological context, low sequence specificity might allow for sequence independent recruitment to blocked replication forks.

Figure: Variable efficacy of binding ability of SLFN11 towards different ssDNA monitored by EMSA.

9) The cryo-EM data collection methods do not contain the number of frames taken per movie. Please add this.

We added the missing information in the cryo-EM methods section (40 frames were taken per movie).

10) ED Fig 6 does not state that C2 symmetry was imposed during refinements. Please add that information.

There was no symmetry imposed during the refinement of the SLFN11^{wt} dimer reconstruction (Supplementary Fig. 8a) as imposition of C2 symmetry did not lead to a significant improvement of the respective reconstruction.

11) The angular distribution plots (and the 25 degree tilt approach) indicate problems with preferred particle orientation. Could the authors provide 3D FSC plots to confirm the isotropy of the resolution they report?

We have added 3D FSC plots for all reconstructions (Supplementary Fig. 8, 9, 10).

Signed review: Christian Dienemann

REVIEWERS' COMMENTS

Reviewer #2 (Remarks to the Author):

The authors have satisfied my concerns.

Reviewer #3 (Remarks to the Author):

Metzner et al have addressed all points appropriately. There is no reason why this manuscript should not be recommended for publication.

We thank the editor and all reviewers for their positive feedback on the revised version of the manuscript. We hope that we address all remaining issues in the final version of the manuscript.

Point-by-point response to referees:

Reviewer #2 (Remarks to the Author):

The authors have satisfied my concerns.

We thank the reviewer for the kind appreciation of our revised manuscript.

Reviewer #3 (Remarks to the Author):

Metzner et al have addressed all points appropriately. There is no reason why this manuscript should not be recommended for publication.

We thank the reviewer for his recommendation of our manuscript for publication.